



# Semi-analytical approach to study the role of abyssal stratification in the propagation of potential vorticity in a four-layer ocean basin

Beatrice Giambenedetti[1,*], Nadia Lo Bue[1,*], and Vincenzo Artale[1, 2,*]

[1]Istituto Nazionale di Geofisica e Vulcanologia (INGV), Via di Vigna Murata 605, 00143 Rome, Italy
[2]National Research Council (CNR), Institute of Marine Sciences (ISMAR), Via del Fosso del Cavaliere 100, 00133 Rome, Italy
[*]These authors contributed equally to this work.

**Correspondence:** Beatrice Giambenedetti (beatrice.giambenedetti@ingv.it)

**Abstract.** Observations of abyssal variability performed in the Ionian Sea (Mediterranean Sea) revealed the presence of a dense stable abyssal layer, whose thermohaline and dynamical properties changed drastically over a decade. Building upon these available observations, we aim to investigate the role that stratification can have on the propagation of vorticity throughout the water column to the abyss, and in turn on the redistribution of the energy stored in the deep sea, with a set of stationary states. A

quasi-geostrophic equation with four coupled layers, a free surface, and a mathematical artifice for parametrizing decadal time evolution has been considered, proving that the relative layer thicknesses and the density difference among the layers are the two critical factors that determine the dynamical characteristics of this propagation. The variability of the ocean stratification is a relevant aspect that can activate the deep and intermediate dynamics engaging in the propagation and stabilization of signals throughout the water column. This demonstrates the non-negligible active connection of the dynamics of the bottom layers with the surface.

The theoretical framework and the parametrization used were based on specific observations made in the Ionian Sea in the last decades, but without losing general applicability in all ocean basins that are characterized by the presence of a stratified dense water mass in the deep and intermediate layers.

## 1 Introduction

The effects of the deep variability on the dynamics at the surface and intermediate depths are generally not well investigated,

despite the growing evidence of the large amount of thermal energy stored in the abyss (Desbruyères et al., 2016; Riser et al., 2016; Meyssignac et al., 2019; Johnson and Doney, 2006; Heuzé et al., 2022). The role of deep dynamics, far from being negligible, in redistributing such energy through small-scale mixing activation is still an open question (Ferrari et al., 2016; De Lavergne et al., 2016; Mashayek et al., 2017b, a; McDougall and Ferrari, 2017).

Given the dynamical characteristics of the Mediterranean Sea, which is generally considered a "miniature ocean" with its

own thermohaline circulation governed by density-driven processes (Bethoux et al., 1999), it turns out to be an ideal lab for experimenting with the impact of stratification in the deep layers processes, with shorter time scales than in the global oceans. In particular, the Ionian Sea located in the eastern part of the Mediterranean Sea, is a perfect site to study abyssal dynamics because it connects two main areas of deep water formation (i.e., the Adriatic and Aegean Seas), and its deep variability is



particularly interesting for ocean circulation (Theocharis et al., 1993; Pinardi et al., 2019; Wüst, 1961; Hainbucher et al., 2006;

Malanotte-Rizzoli et al., 1999). This makes the Ionian Sea representative of ocean basins with deep and intermediate water inputs, giving this case study a more general applicability.

Observations made in the Ionian Sea depths throughout the last 30 years or more, either from deep profiling CTD (Conductivity, Temperature, and Depth) casts, moorings, or seafloor observatories, reveal variability at both interannual and decadal scales, thus showing how active the Mediterranean abyss is (Artale et al., 2018; Hainbucher et al., 2006; Manca et al., 2006;

Budillon et al., 2010), despite having sparse observations and time-series gaps. The dense and stable stratification of the deep layer observed in the Ionian Sea in nearly full-depth profiles could be a key condition for catching deep variability (Giambenedetti et al., 2023). Analysis of different datasets in the deep layers of the Ionian Sea suggested the presence of vorticity likely generated by the stream instability in the deep (Rubino et al., 2012; Giambenedetti et al., 2023). Despite the advancements of technology, the scarcity of available data below $2000m$ of depth remains a challenge that limits our understanding

of the deep sea. To address this gap and gain deeper insights into this complex system, it is crucial to integrate approaches combining existing observations, theory, and numerical models.

To investigate the role that stratification can have on the propagation of perturbation at depth, we started from the hydrological observations of the Ionian Sea. The observed structure suggests that a 4-layer scheme should be enough to have a realistic yet straightforward theoretical representation. This representation adds some complexity to the general view of the Ionian Sea

as a two-layer system (Rubino et al., 2020; Gačić et al., 2021).

Since we want to focus on the abyss response rather than the more active surface and intermediate layers, the quasi-geostrophic (QG) approximation theory is the most suitable for addressing our problem. Hence, we focused on potential vorticity, which can be induced by circulation and gyres, and whose dynamic variability is well known to control flow stability in rotating fluids (Sutyrin, 2015; Dritschel, 1989). The stability of vortices is indeed considered an important topic, as it plays

an active role in the redistribution of the excess heat in the ocean (Wunsch and Ferrari, 2004; Cushman-Roisin, 1987; Armi, 1978; Cessi et al., 2006; Danabasoglu et al., 1994; Abernathey et al., 2010), although many vortices in nature have a longevity that is not correctly represented by theory (Benilov, 2005; Rubino et al., 2007).

Typically, potential vorticity propagation studies are focused on surface and intermediate layers, which have more rapid responses as opposed to abyssal layers and are characterized by the presence of many energetic processes, like for example

thermocline ventilation and diabatic mixing (Smith and Vallis, 2001; Yassin and Griffies, 2022).

The discretization we used for the layered model uses a $z$ vertical coordinate, which is typically used only in the surface layers in numerical models of the ocean, while the interior, which is considered more stable, is treated using a $\rho$ coordinate to follow isopycnals (Griffies et al., 2000). Given the presence of the dense abyssal layer observed in the Ionian Sea, the best representation is depicted by a QG equation with four non-linearly coupled layers having parameters based on in-situ data.

However, considering only four layers results in density differences too big to consider a $\rho$ vertical coordinate in the layered scheme, hence we choose instead the $z$ coordinate. The two schemes are equivalent when $\rho$ can be considered as a linear function of $z$, but this is not our case. However, the presence of high relative density differences in the abyssal layers is not only a local phenomena of the Ionian Sea, in fact this is a wider tendency of the global oceans: ocean stratification has increased





substantially over the past decades, even at great depths (Izquierdo and Mikolajewicz, 2019; Heuzé et al., 2022). Hence, our

study also wants to be indicative of the fact that abyssal stratification should be considered with more attention in models, and neglecting its presence could bring to biased representations of the ocean dynamics.

The approach used is the "equivalent barotropic", which is the simplest one to take into account realistically the effects of stratification (Benzi et al., 1982). To further simplify the model, no external forcing is applied, bathymetry is ignored, and only dissipation is retained, with a small-scale dissipation term and a numerical dumping term in the solution of the QG equations

system to stabilize noise.

Each stratification configuration is solved in time to reach stability, assumed as the state at which dissipation between consecutive iterations become negligible, to obtain a set of stationary states. This can be seen as a time parametrization, meaning that the different thicknesses of the deep layers can be representative of the stratification structures evolving over decades, which can be inferred as the timescales of the deep stratification adjustment observed during the Eastern Mediterranean Transient

(EMT) in the Ionian Sea (Artale et al., 2018; Klein et al., 1999; Malanotte-Rizzoli et al., 1999). The EMT consisted of a transition of deep Ionian water source from the Adriatic Sea to the Aegean Sea, which are warmer and saltier, that lead to a drastic change in the stratification, affecting the Mediterranean Sea circulation over the following decades. This event, that started in the early 1990s, stimulated intense research, but the causes are still debated among the oceanographic community. The EMT is generally linked to surface changes (Borzelli et al., 2009; Pinardi et al., 2019; Amitai et al., 2017) impacting the

deep, not considering the other way around, or the concurrence of deep and surface mechanisms in the process.

The mathematical artifice we decided to implement to parametrize the different time scales involved in the abyssal stratification changes, is a way of looking at the complexity of the connection between surface and depth with an upside-down point of view, aiming to explain the observed variability in the Ionian abyssal layers.

## 2    Observations and problem formulation

In Giambenedetti et al. (2023) is shown the presence of a deep dense stable layer in the Ionian Sea, and an enhancement of diffusivity in the abyssal layer due to tides. The presence of this stable layer has been suggested to be a key factor for observing the mixing activation in the abyss. The dataset presented in Giambenedetti et al. (2023) has been integrated here with long-time series of CTD and current meter from the NEMO-SN1 multidisciplinary seafloor observatory.

The NEMO-SN1 observatory is located in the Western Ionian Sea off Eastern Sicily (Italy), about $25\,km$ from the coast

($37.5°N, 15.4°E$), at a depth of $2100\,m$, and the two missions to which this study refers were carried out from October 2002 to February 2003 (named GNDT-1) and from June 2012 to June 2013 (named SMO-1) (Favali et al., 2013; Lo Bue et al., 2019). Both the observatory data and the Giambenedetti et al. (2023) CTD survey data show the presence of the dense stable layer in the same period. Figures 1a and b clearly show how the thermohaline and currents properties of the bottom layers of the Ionian Sea have drastically changed over 10 years. In fact, the potential density anomaly with reference pressure of

$2000\,dbar$ $\sigma_2$ undergoes a change of $\Delta\sigma_2 = 0.05\,kg/m^3$ (colorbar in Figures 1a and b), which is four times bigger than the usual range of inter-annual variability at these depths. Moreover, the currents underwent drastic changes: even though



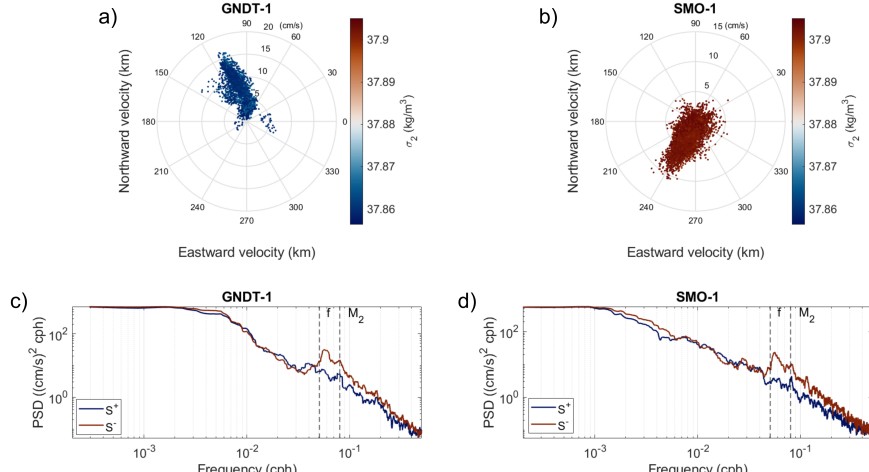

**Figure 1.** Current hodographs of (a) 2002-2003 (GNDT-1) and (b) 2012-2013 (SMO-1) missions from NEMO-SN1 observatory, where the colorbar indicates the density anomaly ($\sigma_2$) values (Lilly and Rhines, 2002). Rotary spectra of (c) 2002-2003 (GNDT-1) and (d) 2012-2013 (SMO-1) current meter data from NEMO-SN1 observatory, where color refers to the clockwise/anticlockwise spectra (S±); dashed grey lines indicate the inertial ($f$) and tidal ($M_2$) frequencies.

the current's amplitude has a similar range of variability, the current's direction changed completely. An interesting feature captured by the NEMO-SN1 observatory current time series can be seen in the rotary spectra in Figures 1c and d, hinting at the presence of vorticity in the abyss. In fact, the spectra are dominated by inertial oscillations at the Coriolis frequency ($f$), "blue

shifted" from the local Coriolis frequency, i.e. shifted towards higher frequencies by the presence of a background potential vorticity (Kunze, 1985). The near-inertial peak for 2002-2003 mission (GNDT-1) is $f_{GNDT-1} = 0.0577\,cph$ (Fig. 1c) and for 2012-2013 mission (SMO-1) $f_{SMO-1} = 0.0562\,cph$ (Fig. 1d) i.e., there is a blue shift of $\sim 12 - 10\%$ respectively, which is consistent with the frequency excess of $10\%$ in the inertial band reported in literature (Garrett, 2001). The properties of the internal waves are significantly dependent on the stratification condition of the area, which is straightforward in shallow waters

(Kurkina et al., 2017; van Haren, 2015). However, even at great depth, in which the change in the thermohaline properties happens over decades, the bottom layers stratification variability can impact the properties, propagation, and interactions of near inertial waves.

Those hints, combined with the findings of Giambenedetti et al. (2023), are the starting point to the development of our idealized model, hence the model parameters are chosen based on the nearly full depth CTD profiles used in Giambenedetti

et al. (2023). The surveys were performed between 1999 and 2003 in the Ionian Sea ($36°18' N, 16°6' E$). Therefore, the Coriolis parameter was computed at the sampling site coordinates, the layer thicknesses were chosen from the water masses defined by the Temperature and Salinity profiles (Figure 2a). The densities were computed following the international standards with the TEOS-10 routine (https://teos-10.org/) within each layer (Figure 2b) and kept constant for all configurations so that there is always a $4^{th}$ denser layer.



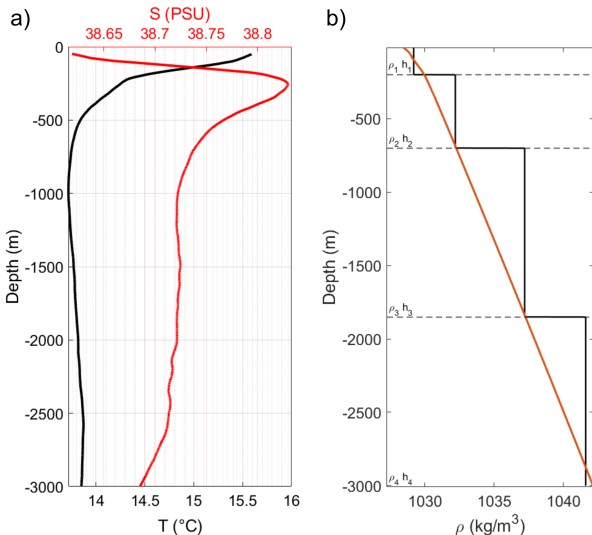

**Figure 2.** (a) Temperature and Salinity profiles performed during April 2002 in the Ionian Sea, from the dataset https://doi.org/10.5281/zenodo.7871734; (b) average vertical density profile of the in-situ reference data (solid orange line) and its piecewise constant approximation (solid black line). Dashed lines show interfaces between layers, in this case $h_3 = h_4 = 1150\,m$.

Especially at depth, it can be seen in Figure 2a how the thermohaline properties of Ionian Sea waters are more salinity-driven, since the temperature profile is constant in the bottom layers. Figure 2b gives an example of the piecewise constant density used for one of the configurations compared with the reference in situ density profile, in particular, we set the densities of the four layers to be: $\rho_1 = 1029.2\,kg/m^3$, $\rho_2 = 1032.2\,kg/m^3$, $\rho_3 = 1037.2\,kg/m^3$, $\rho_4 = 1041.6\,kg/m^3$. The Brunt-Väisälä frequency has been computed from the density profiles and corrected for compressibility, using the TEOS-10 routines as well.

## 3   Methods

Starting from the QG equations for potential vorticity for nonlinear motions in a continuously stratified fluid on a beta plane (Pedlosky, 1987; Vallis, 2017; Cushman-Roisin and Beckers, 2011):

$$\frac{\partial q}{\partial t} + J(\psi, q) = 0 \tag{1}$$

where $J(\cdot, \cdot)$ stands for the Jacobian operator, the pressure field is transformed into a streamfunction via $p = \rho_0 f_0 \psi$, $f = f_0 + \beta y$

is the Coriolis parameter in beta-plane approximation, and the potential vorticity is defined as:

$$q = \nabla^2 \psi + \frac{\partial}{\partial z}\left(\frac{f_0^2}{N^2}\frac{\partial \psi}{\partial z}\right) + \beta_0 y \tag{2}$$

This can be seen as a generalization of Ertel's theorem in hydrostatic approximation with a different formulation (McWilliams, 2006).





To describe the 4-layer, two-dimensional (2D) ocean we consider arbitrary adimensional thicknesses $h_j^* = h_j/H$ with $j = 1, ..., 4$ and $H$ the vertical scale, and constant adimensional densities within the layers $\rho_j^* = \rho_j/\rho_0$ with $j = 1, ..., 4$ (see Figure 2). The dynamics are described in each layer by the streamfunction $\psi$ and the potential vorticity $q$ (eq. (1) and (2)).

Equation (2) contains derivatives in $z$, which must be discretized to conform with a $4-$layer representation. We use a straightforward finite-difference technique for discretization, considering fixed levels in $z$ to define the layers instead of density. Using $z$ as a vertical coordinate makes it possible to use the simplest discretization approach, accurately representing pressure gradients and equation of state for a Boussinesq fluid (Griffies et al., 2000). Since the discretization is performed on $z$ directly, instead of taking $\rho$ as the vertical variable, the formalization is slightly different than usual QG layered models (Vallis, 2017; Cushman-Roisin and Beckers, 2011). The two approaches are equivalent when considering long-scale systems where density is a linear function of $z$. In global models, they are both used and combined also with a $\sigma$-coordinate for the bottom boundary layer, depending on the ocean region considered and the numerical cost. Still, since we seek stationary states parametrizing a long-term evolution of the layers, the difference between choosing $\rho$ or $z$ as the vertical variable is infinitesimally small and simplifies the mathematical treatment.

Because of layers' uniform density and fixed arbitrary nonzero thicknesses, we have four immiscible, inviscid fluids of different densities contained between interfaces (Flierl, 1978). This means that the stratification frequency in (2) is uniform within the layers. Hence, given the piece-wise constant definition of the density (Figure 2), and the buoyancy frequency definition, we can define $N^2$ (for the $j-th$ layer, with $j = 1, ...4$) as (Sokolovskiy, 1997; Cushman-Roisin and Beckers, 2011):

$$N_j^2 = -\frac{g}{h_j}\Delta\rho_j \tag{3}$$

where $\Delta\rho_j = \frac{(\rho_j - \rho_{j-1})}{\rho_0}$, and $g\Delta\rho_j = g_j'$ is the reduced gravity.

The boundary conditions we considered in the vertical are flat bottom, which require zero vertical velocity at $z = -H$ and translates into:

$$\partial\psi/\partial z = 0 \tag{4}$$

and free surface at $z = 0$, with no inflow/outflow:

$$\frac{\partial\psi}{\partial z} + \frac{N^2}{g}\psi = 0 \tag{5}$$

The free surface boundary condition is not typically used in layered models, because with a rigid lid the coupling coefficient matrix can be more easily treated (Vallis, 2017; Sokolovskiy, 1997; Carton et al., 2014), but it is a necessary condition for our aims to include the surface response and let it be free of moving. In fact, although we will keep the first two layers thickness configuration fixed, a kinematic condition on the free surface is necessary since we are interested in the vertical propagation.





In discretized form, the boundary condition at the surface can be written introducing an artificial streamfunction for the atmosphere $\psi_0$ and using eq. (3):

$$\frac{\psi_1 - \psi_0}{h_1} + \frac{N_1^2}{g}\psi_1 = 0$$

$$N_1^2 = -g\frac{\Delta\rho_1}{h_1} = -\frac{g}{h_1}\left(\frac{\rho_1}{\rho_0} - 1\right)$$

$$\Rightarrow \quad \psi_0 = \psi_1\left(2 - \frac{\rho_1}{\rho_0}\right) = \kappa\psi_1 \tag{6}$$

when $\kappa = 1$, i.e. $\rho_1 = \rho_0$, one obtains the rigid lid condition (Sokolovskiy, 1997). The boundary condition at the bottom can be simply set as: $\psi_5 = \psi_4$, where $\psi_5$ is an artificial streamfunction for the seafloor.

The streamfunctions in the layers then are defined independently, and are connected at the interface through the finite difference coming from the second derivatives in (2):

$$\left.\frac{\partial^2\psi}{\partial z^2}\right|_1 \simeq \frac{\psi_0 - 2\psi_1 + \psi_2}{\Delta z^2} = \frac{(\psi_2 - \rho_1\psi_1)}{h_1^2} \quad \text{(Rigid Lid for } \rho_1 = 1) \tag{7a}$$


$$\left.\frac{\partial^2\psi}{\partial z^2}\right|_2 \simeq \frac{\psi_1 - 2\psi_2 + \psi_3}{\Delta z^2} = \frac{(\psi_1 - 2\psi_2 + \psi_3)}{h_2^2} \tag{7b}$$

$$\left.\frac{\partial^2\psi}{\partial z^2}\right|_3 \simeq \frac{\psi_2 - 2\psi_3 + \psi_4}{\Delta z^2} = \frac{(\psi_2 - 2\psi_3 + \psi_4)}{h_3^2} \tag{7c}$$

$$\left.\frac{\partial^2\psi}{\partial z^2}\right|_4 \simeq \frac{\psi_3 - 2\psi_4 + \psi_5}{\Delta z^2} = \frac{(\psi_3 - \psi_4)}{h_4^2} \tag{7d}$$

where we used a straightforward central finite difference technique, given that it is a second-order derivative and $h_j$ are the adimensional thicknesses. We would like to stress here that this is the crucial point to our choice on vertical coordinate: if this discretization was made on $\rho$ instead of $z$ the finite differences would have been too big to give a good approximation of a derivative in our case. This could be of course worked around adding more levels, but this would lead to a higher computational 170 cost, hence this was the best compromise for our aims.

The set of governing equations can now be written in vectorial form for all four layers as:

$$\frac{\partial \boldsymbol{q}}{\partial t} + J(\boldsymbol{\psi}, \boldsymbol{q}) = 0 \tag{8a}$$

$$\boldsymbol{q} = \nabla^2\boldsymbol{\psi} + \boldsymbol{A}\boldsymbol{\psi} + \beta_0 y \tag{8b}$$





where:

$$\boldsymbol{A} = -f_0^2 \begin{pmatrix} -\frac{\rho_1}{\rho_0}\frac{1}{h_1 g_1'} & \frac{1}{h_1 g_1'} & 0 & 0 \\ \frac{1}{h_2 g_2'} & -\frac{2}{h_2 g_2'} & \frac{1}{h_2 g_2'} & 0 \\ 0 & \frac{1}{h_3 g_3'} & -\frac{2}{h_3 g_3'} & \frac{1}{h_3 g_3'} \\ 0 & 0 & \frac{1}{h_4 g_4'} & -\frac{1}{h_4 g_4'} \end{pmatrix}$$

is the coupling matrix, and:

$$\boldsymbol{\psi} = \begin{pmatrix} \psi_1 \\ \psi_2 \\ \psi_3 \\ \psi_4 \end{pmatrix}$$

is the streamfunction vector for the four layers.

So this term $A$, which is a linear version of vertical stretching, holds in it the non-linear connection between the stratification and the layers' thicknesses, which is now limited at the interfaces. Since the coupling matrix $A$ does not vary with time (densities and thicknesses of the layers are fixed), the system of equations for the layers eq. (8) can be decoupled using a linear transformation approach (Carr, 2021; Sokolovskiy, 1997). Let's introduce $\tilde{\psi} = B^{-1}\psi$ where $B$ is an invertible matrix ($n \times n$ where $n$ is the number of layers involved) such that $B^{-1}AB = \Lambda = diag(\lambda_1,...,\lambda_n)$ are the eigenvalues of $A$. An appropriate

choice is then to take the corresponding eigenvectors as the column of $B$. Multiplying both sides of eq. (8) by $B^{-1}$ the time evolution for each layer is given by:

$$\tilde{q}_n = \nabla^2 \tilde{\psi}_n + \lambda_n \tilde{\psi}_n + B_{n,n}^{-1}\beta_0 y \tag{9a}$$

$$\frac{\partial \tilde{q}_n}{\partial t} + J(\tilde{\psi}_n, \tilde{q}_n) = 0 \tag{9b}$$

for each $n$ layer independently. The Jacobian operator was discretized using the Arakawa scheme (Mesinger and Arakawa, 1976) that ensures conservation laws, i.e. the null integral over a domain with uniform $\psi$ along the boundary, which can be related to the evolution of kinetic energy, and also the anti-symmetry property of the Jacobian $J(\psi, q) = -J(q, \psi)$.

The numerical method used to solve is equivalent to a Runge-Kutta of the $4^{th}$ order (Dormand and Prince, 1980), and the streamfunction is adjourned at each step extrapolating from the previous values, i.e. inverting the Laplacian by applying a

Gauss-Seidel approach.

If turbulent dissipation is retained in the formalism, eq. (1) become more complicated, but for numerical application a good approximation is given by (Cushman-Roisin and Beckers, 2011):

$$\frac{\partial q}{\partial t} + J(\psi, q) = A_H \nabla^2 q \tag{10}$$



where $q$ remains defined by eq. (2), and $A_H$ is a constant for physical dissipation. It can be useful to add an extra term to filter spurious oscillation in the solution due to the aliasing problems created by non-linearities, like a scale-selective biharmonic dissipation of vorticity:

$$\frac{\partial q}{\partial t} + J(\psi, q) = A_H \nabla^2 q - B_H \nabla^2(\nabla^2 q) \tag{11}$$

where $B_H$ controls the damping, and since it is a filter for numerical purposes depends on the grid spacing and time step by construction.

We setted the physical diffusion coefficient to $A_H = 0.1$ and the biharmonic term coefficient $B_H = 10 \cdot \Delta x \Delta y$. The horizontal grid used was $250 \times 250$ for a vortex scale of $L = 8000\,m$. The timescale chosen was $T = 100000\,s$, based on algorithm stability checks made for the case of two layers with perturbed streamfunction, and the time step was constrained by the CFL (Courant-Friedrichs-Lewy) condition (Cushman-Roisin and Beckers, 2011).

For the numerical simulation, we kept the thickness of the first and second layers always fixed at $200\,m$ and $500\,m$ respectively, while changing the relative thicknesses of the third and fourth layers, for a water column of $3000\,m$ in total. This is a reasonable assumption since we are considering long-term effects, neglecting seasonal variability. The $3^{rd}$ layer was then varied from $300\,m$ to $2000\,m$ with a $\sim 50\,m$ spacing, and the $4^{th}$ layer accordingly, for a total of 35 configurations. To observe better the critical behavior, a few more configurations have been added.

The mean flow considered for the basic states is circular (Carton et al., 2014; Katsman et al., 2003; Sokolovskiy, 1997), i.e. a cyclonic vortex centered at the origin with its radial streamfunction given by:

$$\bar{\psi}_j = U_j L e^{-r^2/L} \tag{12}$$

where $r = \sqrt{x^2 + y^2}$ is the radius in cylindrical coordinates, $L$ the horizontal scale of the vortex, and $U_j$ are the layer-wise velocities set to be $U_1 = 1$ and zero for the other layers at the starting point. Figure 3a shows the radial profiles of the layer-wise streamfunction of the initial states. They were chosen as a Gaussian radial profile, consistent with the typical intermediate water eddies/cyclones observed in the Mediterranean Sea (Paillet et al., 2002). More importantly, for a Gaussian vortex, the baroclinic instability has been proven to be dominant (Carton et al., 1989). In the final states, regardless of the configuration, since the stratification coupling effect is a higher-order nonlinearity, the first two layers have always the same behavior (Figure 3b), with small peripheric ringlets, while the other layers' response is small in intensity but significantly higher than numerical noise ($O(10^{-16})$ in our case), and highly dependent on the thickness configuration (Figure 3c).

## 3.1 Qualitative validation

From layer thicknesses, Coriolis parameter, and densities is possible to evaluate the interfacial internal deformation radius $L_d^j = \sqrt{g_j' h_j}/f$, which is a baroclinic Rossby radius of deformation, taking into account the reduced gravity and relative thickness of the layers (LeBlond and Mysak, 1978). The first layer has $L_d \simeq 15\,km$ while the second layer $L_d \simeq 44\,km$, consistent with the typical internal radius values for the first oscillation modes reported by literature (Carton et al., 2014; Nittis et al., 1993).





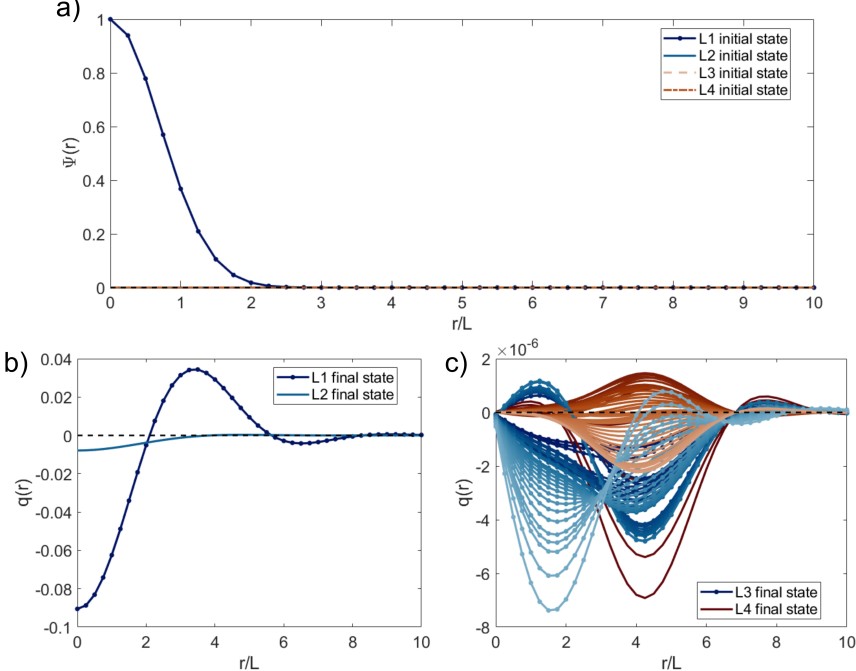

**Figure 3.** (a) Radial profiles of the initial states of streamfunction for the four layers ($L_i$, $i = 1, ..., 4$). Radial profiles of the final states of potential vorticity for (b) the first two layers and (c) the other layers for all different configurations, colored in shades of blue for the $3^{rd}$ and in shades of red for the $4^{th}$ layer.

The $3^{rd}$ and $4^{th}$ layers radii are $O(40 - 120km)$, as shown in Figure 4a, peaking at $\sim 90km$ which conform with observations in the Ionian Sea in which the only persisting scales are ascribable to gyres at intermediate and deep levels (Nittis et al., 1993; Robinson et al., 1991; Pinardi and Masetti, 2000). Of course, these high values are dependent on the abrupt density variation considered using the piecewise constant density in the model (Figure 2b). However, falling in the same order of magnitude of different observations can be a hint of the fact that we are not departing too much from reality.

To overcome the data paucity, exploiting the available resources and former observations, the only parameter we found able to give a reasonable comparison of very different observations with our model data is the phase speed of the internal modes (LeBlond and Mysak, 1978). In Figure 4b there is a comparison of phase speed values of our model with two available observations performed in the Ionian Sea deep layer: one is the abyssal vortex identified by Rubino et al. (2012), and the other is the blue-shift peak observed in the currents spectrum of NEMO-SN1 observatory in the same period (Fig. 1a). The values

of the phase speed from our model take into account both the scales of the parameters and the timescale of the integration to get a more realistic speed value ($v_p = \sqrt{g'_j h_j} \cdot (H/LTf_0)$). The first layer has $v_p \simeq 5.86 cm/s$, the second layer $v_p \simeq 16.4 cm/s$, the $3^{rd}$ layer phase speeds are $O(10 - 50 cm/s)$, and for the $4^{th}$ layer $O(10 - 40 cm/s)$. The estimates from Rubino et al. (2012) take into account the background flow (which is opposite the traveling direction of the vortices depicted in Figure 3a of Rubino et al. (2012) and the Rossby radius in Table 1 in their paper to have a scaling factor comparable to our



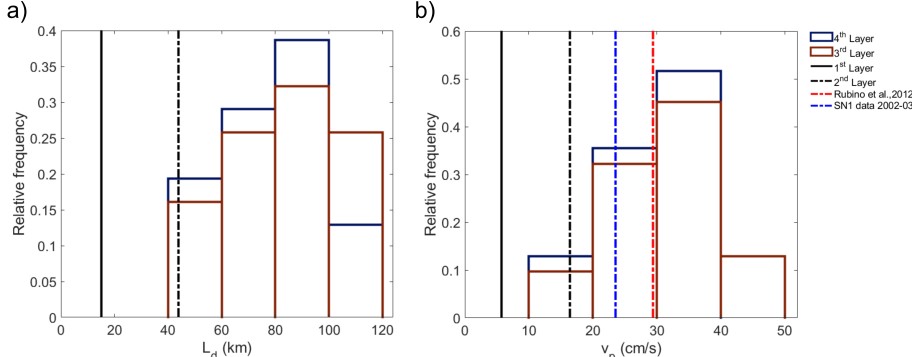

**Figure 4.** (a) Relative frequency of internal deformation radii values for the different configurations of the $3^{rd}$ and $4^{th}$ layer. The values of the $1^{st}$ and $2^{nd}$ layers radii are also reported. (b) Relative frequency of Rossby phase speed values for the different configurations of the $3^{rd}$ and $4^{th}$ layer. The values of the $1^{st}$ and $2^{nd}$ layers speeds are also reported in black, as well as estimates of phase speed for observed vortices in the Ionian Sea by Rubino et al. (2012) (red) and for current time series from the NEMO-SN1 observatory (blue).

$(v_p = (\lambda/T + bkg) \cdot (H/Rr_{vortex}) \simeq 29.4\,cm/s$ where $\lambda$ is the wavelength and $T$ is the period of the first anticyclone, $bkg$ stands for background). The NEMO-SN1 phase speed estimate is based on the current spectra blue-shifted peak in Figure 1a $(v_p = \omega/k \simeq 23.5\,cm/s$ i.e., frequency over wavenumber).

## 4   Results and discussion

The potential vorticity that propagates through the layers significantly differs when the relative thicknesses of the bottom layers

are altered (Figure 3c). If the relative thickness of the different layers had a negligible effect for all the possible configurations, according to the design of our numerical model, no changes should occur and the potential vorticity would remain confined in its original state. However, the second order stretching term (eq. (8)b), connecting the water column through the density difference among the layers and their different thicknesses, seems to have a bigger impact for some configurations of the $3^{rd}$ and $4^{th}$ relative thicknesses.

To investigate if there were any typical configurations, we plotted the final states in the $(q_i, q_{i+1})$ parameter space $(i = 1, 3)$, chosen because layer potential vorticities directly affect the system stability (Bashmachnikov et al., 2017), building some sort of stability phase diagram, which however resulted to be quite complex (Figures 5a and b). In Figure 5a the final potential vorticity of the two first layers, which are never perturbed, is not exactly the same for all configurations. There is a shift for some of them, while the overall pattern remains the same (i.e., the radial distribution like in Figure 3). This suggests that changing the

configurations of the $3^{rd}$ and $4^{th}$ layer thicknesses influences the transmission of potential vorticity through the bottom layers (and vice versa). In particular, there is more transmission for $h_3/h_4 = 0.74, 0.77, 1, 1.1, 1.2, 1.3, 1.4, 1.5, 1.6, 3.6, 4.1, 4.4$ i.e., the 'bumps' in Figure 5a, in correspondence of which there are some bigger features in the bottom layers potential vorticity (Figure 5b).



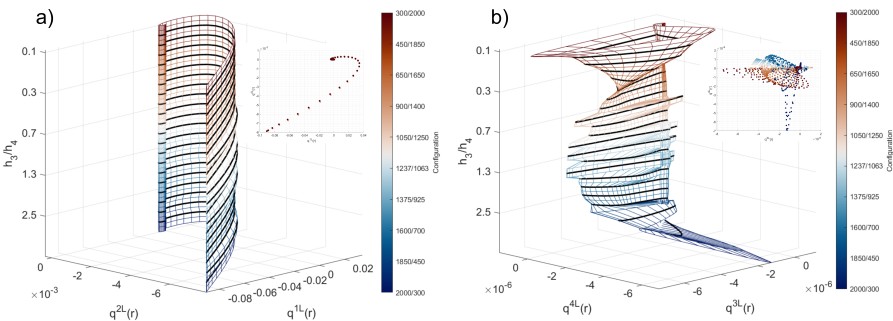

**Figure 5.** Final state potential vorticity in $(q_i, q_{i+1})$ parameter space ($i = 1, 3$) for (a) $1^{st}$ and $2^{nd}$ layers, and (b) $3^{rd}$ and $4^{th}$ layers meshed with relative thickness in the vertical (the ticks are equispaced for readability since there are some extra configurations), contours are colored with the configurations of bottom layers' thickness. The smaller panels in the right upper corners show the same in 2D. The plotting angle chosen makes evident the inversion nodes for $h_3/h_4 = 0.67$ and $h_3/h_4 = 4.1$.

In Figure 5b it can be seen how the potential vorticity of the bottom layers changes drastically in response to varying

thicknesses, without showing a regularity. However, in the 2D plot panel, a trend can be observed where the vorticity of the $3^{rd}$ and $4^{th}$ layers tends to align with the spiral pattern of the first two layers when the first and last configurations are applied (dark blue and dark red). This tendency is disrupted by the non-linearity as we get close to $h3/h4 = 1$, which corresponds to the 'bumps' mentioned earlier in the first two layers. This means that the strong non-linear coupling of the deep layers' relative thickness and density affects the whole water column.

From the spatial fields of limit values of $h_3/h_4$, i.e., the lower, highest, and about-critical values, it can be seen how vorticity penetrates more in the $3^{rd}$ layer when $h_3/h_4 << 1$ (Figure 6a) and more in the $4^{th}$ when $h_3/h_4 >> 1$ (Figure 6 d). Before the threshold transition, the $3^{rd}$ layer has higher potential vorticity values than the $4^{th}$ (Figure 6b), after the transition the potential vorticity in the $3^{rd}$ and $4^{th}$ layers is almost the same (Figure 6c), and it can be seen in particular looking at the outer corners of the vortex in the $3^{rd}$ layer, how the rotational direction inverts (Figure 6b and c).

This behavior is not reproduced by the 2-layer (Figure 7) and 3-layer (Figure 8) cases, simulated under the same conditions as the 4-layer case and for the same thickness ratios of bottom layers. In Figures 7 and 8 can be seen how the coupling has the effect of slowly passing more potential vorticity to the bottom layer.

When exploring the modal projection of the streamfunction the impact of varying stratification configurations on the system becomes even more apparent (Figure 9). The modes are computed from eq. (8) when $q = 0$ via separation of variables, and

then the evolved streamfunction is projected on the eigenvector basis (Smith and Vallis, 2001; Katsman et al., 2003). There is a conversion between the $1^{st}$ and $2^{nd}$ baroclinic modes that alters the sign of the barotropic mode for the value before the critical transition (Figures 9a and b).The $1^{st}$ and $2^{nd}$ modes are the more impacted by the stratification configuration. The $1^{st}$ mode has more modal energy near the transition (Figures 9b and c), while the $2^{nd}$ is more energetic away from the transition




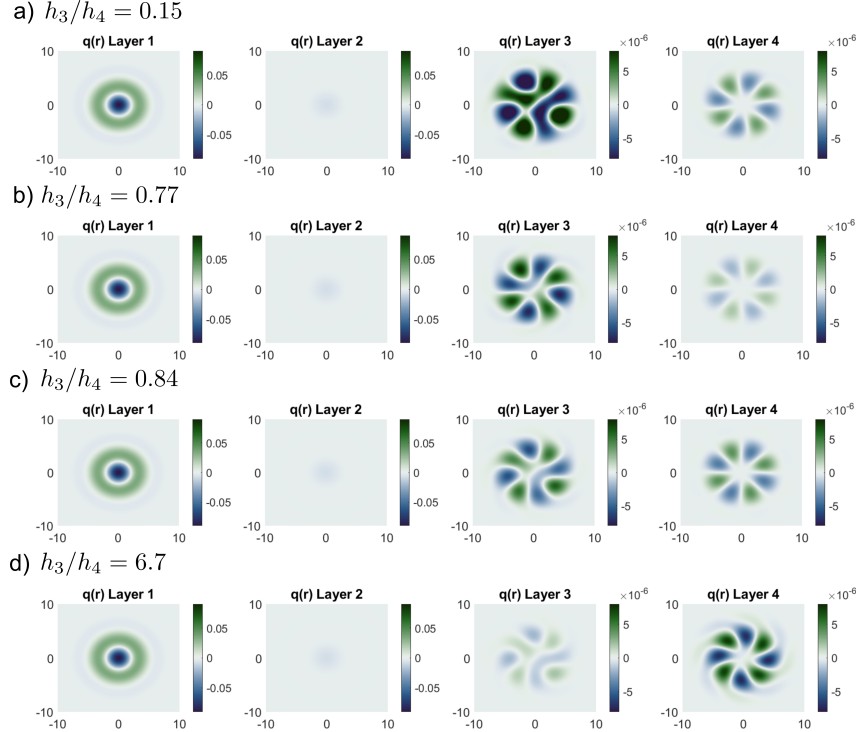

**Figure 6.** Spatial fields of potential vorticity in the 4-layer case for (a) $h_3/h_4 = 0.15$, (b) $h_3/h_4 = 0.77$, (c) $h_3/h_4 = 0.84$, and (d) $h_3/h_4 = 6.7$. The colorbar ranges are different for different layers.

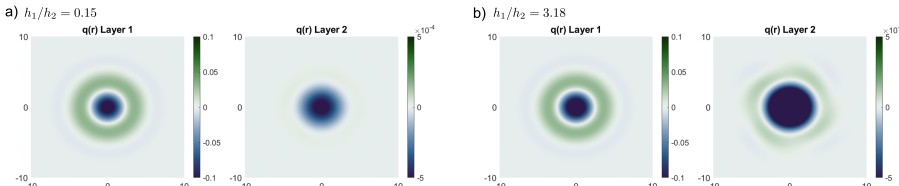

**Figure 7.** Spatial fields of potential vorticity in the 2-layer case for (a) $h_1/h_2 = 0.15$ and (b) $h_1/h_2 = 3.18$. The colorbar ranges are different for different layers.

(Figures 9a and d) and with different signs. On the other hand, the $3^{rd}$ mode is almost identical for all the configurations, with
slightly more or less modal energy away from transition (Figures 9a and d).

This critical behaviour becomes more evident when choosing a more appropriate variable to investigate the non-linearity region, i.e. the ratios between $3^{rd}$ and $4^{th}$ layers' vorticity and the correspondent thickness ratio (Figure 10). In particular, Figure 10a shows the ratio of the sign-preserving max absolute values of $3^{rd}$ and $4^{th}$ layers potential vorticities, which can be





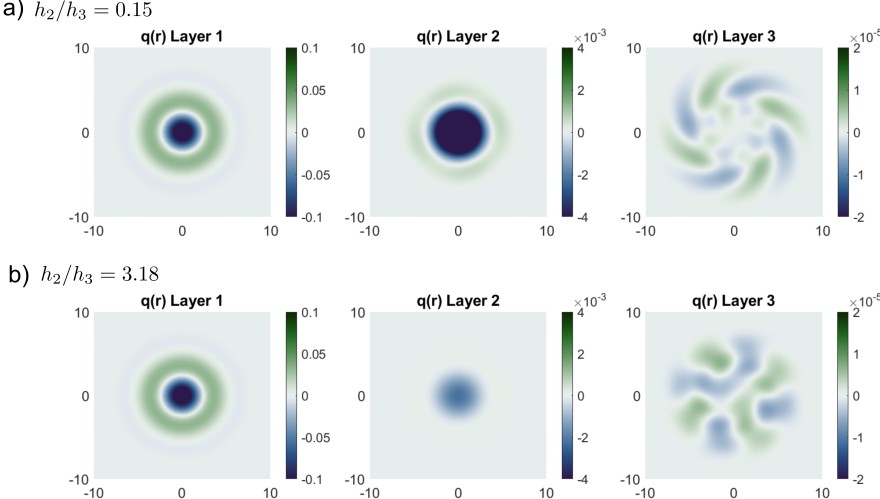

**Figure 8.** Spatial fields of potential vorticity in the 3-layer case for (a)$h_2/h_3 = 0.15$ and (b) $h_2/h_3 = 3.18$. The colorbar ranges are different for different layers.

defined for a generic variable $x$ as:

$$sign(min(x) + max(x)) \cdot max(|min(x)|, |max(x)|) \tag{13}$$

As the thickness of the $3^{rd}$ layer increases (and subsequently the thickness of the $4^{th}$ layer decreases), the vorticity signal initially tends to rotate in same direction as long as it reaches a critical ratio below $h_3/h_4 = 1$ where a sort of phase transition occurs (Parisi and Shankar, 1988). This behavior is not present in either the 3-layer case (Figure 10b) or the 2-layer case (Figure 10c), where the potential vorticity slowly passes more to the bottom layer monotonically.

In particular, the critical ratio found for the 4-layer case is $\sim 0.77$, which is close to the $0.8$ value where marginal stability curves meet in the two-layer case, as found by Benilov (2003). This is also the configuration for which the barotropic mode streamfunction changes sign (Figure 9b). Finding the presence of such critical configurations is a retrospective justification for the time parametrization used (i.e., exploring the stable configuration independently on time evolution) since inside an evolutionary model this would lead to a non-convergence of the algorithm. The potential vorticity signal thus increases in the $4^{th}$ layer and the rotation direction inverts. Therefore, regardless of stronger dissipation, forcing, and other instabilities one considers in a more accurate description of the flow dynamics, the net effect of the deep stratification variability is to reshape the structure of the mean circulation and mass distributions throughout the whole water column.

This stratification-driven reshaping can have important implications for understanding realistic vortex stability: it has been suggested that a weak co-rotating circulation in the lower layer of a vortex can stabilize it, thus being an eligible explanation for the observed longevity of oceanic vortices (Dewar and Killworth, 1995; Benilov, 2005). Moreover, Benilov (2005) shows that instead of considering the deep circulation to have the same profile as the upper layer vortex (like in Dewar and Killworth (1995) and Katsman et al. (2003)), it is more realistic to consider constant potential vorticity in the deep. This implies that



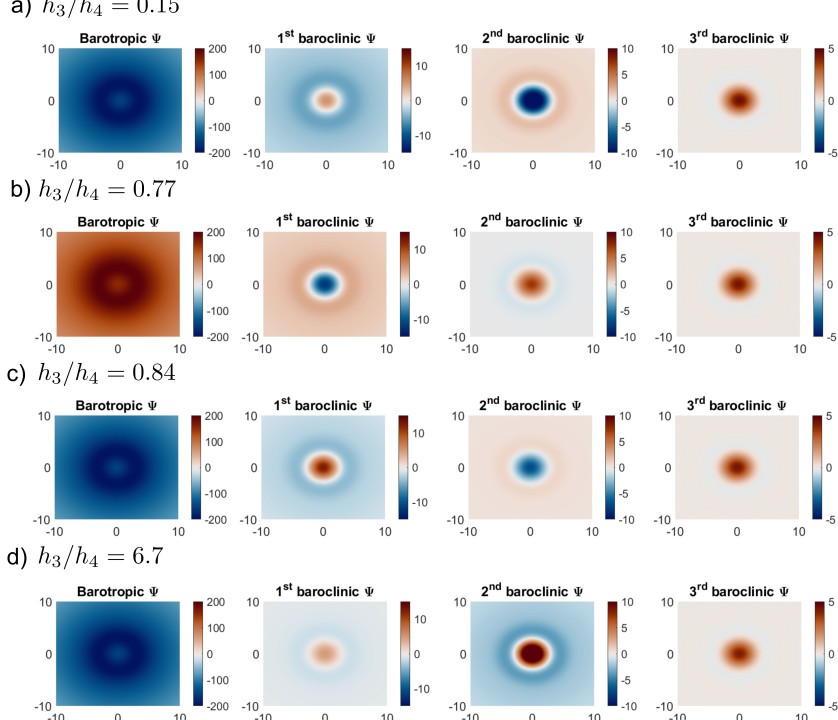

**Figure 9.** Streamfunction projection on the barotropic and baroclinic modes in the physical plane for (a) $h_3/h_4 = 0.15$, (b) $h_3/h_4 = 0.77$, (c) $h_3/h_4 = 0.84$, and (d) $h_3/h_4 = 6.7$. The colorbar ranges are different for different modes.

the resulting circulation from the upper layer displacement is always co-rotating, hence acting to stabilize it. Alternatively, as proposed by Sutyrin (2015), the presence of a middle layer with uniform potential vorticity can weaken vertical coupling and
enhance vortex stability.

To explore the stability of the system (which in our case is a small order effect, but is still worth the attention), we used the Benilov (2003) definition for vortex stability inside the $i-th$ layer, which is given by the sign of the potential vorticity radial derivative with respect to the azimuthal velocity of the vortex $dQ_i/dr \rightarrow -V_i$. Figure 11 shows the radial derivatives of potential vorticity for all the layers. For the first two layers (Figures 11a and b) also the radial derivative of streamfunction, that
is the radial velocity. However, this latter cannot be shown on the same plot for the other layers for readability, as it depends on the configuration as well (Figures 11c and d).

The vortex in the first and second layers (Figures 11a and b) is stable at the center and becomes unstable at the borders, as it should be for dissipation. Overall for the $3^{rd}$ and $4^{th}$ layers the trend is very different and not comparable, except for some configurations. In particular, configurations with smaller $3^{rd}$ layer (smaller than $450\,m$) have the same sign (very unstable vor-
tex), and configurations with $h_3 = 975, 1000, 1150, 1200, 1250, 1300, 1350, 1375, 1400, 1800, 1850, 1875$ have opposite signs (very stable vortex). Even in the absence of disturbances, the equilibrium state is not stable for all the configurations of layers.



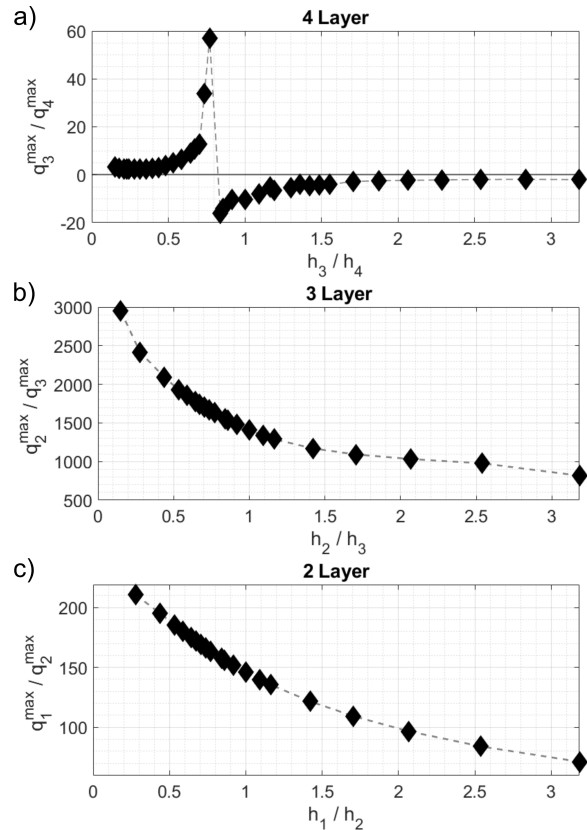

**Figure 10.** The ratio of the sign-preserving max absolute values of $(n-1)-th$ and $n-th$ layers potential vorticities versus thicknesses ratio for (a) 4-layer case, (b) 3-layer case, and (c) 2-layer case.

For what concerns the overall stability of the vortex throughout the water column, one can consider a similar criterion to that of Rayleigh's inflection point, i.e. the vortex is stable when $dQ_i/dr$ does not change the sign in the other layer (McWilliams, 2006). Comparing the signs of $dQ_i/dr$ among the layers (Figure 11), it can be seen how changing the deep stratification can influence the stability of the potential vorticity profiles, particularly for the second layer with respect to the $3^{rd}$, which in turn is strongly coupled with the $4^{th}$. This of course is a reflection of the change in the potential vorticity sense of rotation highlighted before.

The fact that the layers coupling through stratification instantaneously creates a circulation pattern in the layers below with a rotation sign related to the relative thickness of the layers could be an important factor that affects the stability/instability of vortices. In a QG framework, when considering 'ocean-like' stratifications, i.e. considering the upper layers pycnocline but an abyss at rest, the equilibration mechanisms for vortexes are typically set a priori (Smith and Vallis, 2001). Our results suggest that adding a little complexity to the vertical structure of the abyssal layers could give a more realistic picture without setting a priori conditions for stability. However, the contribution of the deep stratification we found is quite small when comparing the





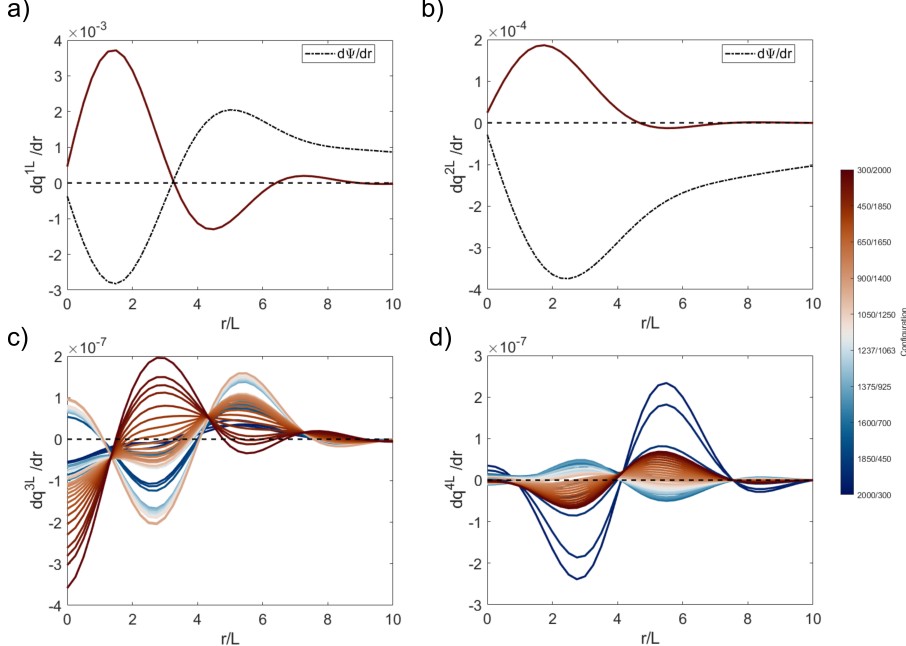

**Figure 11.** Radial derivative of potential vorticity for (a) $1^{st}$, (b) $2^{nd}$, (c) $3^{rd}$, and (d) $4^{th}$ layer. Dashed lines indicate the zero line, and the dashed-pointed line in (a) and (b) shows the radial derivative of the streamfunction.

potential vorticity values in the upper layers with the below ones. To further test the implications and the possible impacts, we
considered the vortex evolution under the same conditions, introducing a periodic radial perturbation in the streamfunction (eq. (12)). This is defined as $r = r \cdot (1 + \epsilon \cos 2\theta)$, where $\theta$ is the azimuthal angle and $\epsilon$ a small parameter so to have a variation of $\psi$ with a finite but small amplitude. We consider $2\theta$, since in a single-layer case this choice would lead to a known case, with a dipolar breaking of the original vortex patch, so to have a reference to interpret the results in the 4-layer case (Carton et al., 2014; Cushman-Roisin and Beckers, 2011).

In Figures 12 and 13 are shown the vortex patches evolutions for bottom layers thickness ratio of $h_3/h_4 = 0.15$ and $h_3/h_4 = 0.77$ respectively. The overall evolution of the first layer is different from the expected one, the presence of the layers below of course generates some effects through potential vorticity conservation, but with respect to the non-perturbed case these effects are greater, both in terms of instability of the patch evolution geometry and potential vorticity magnitude. As seen also in the non-perturbed case, the non-linear coupling creates automatically a non-zero vorticity in the second layer, which then
propagates to the bottom layers. The evolution of a vortex interacting with a vortex patch in the middle layer has been studied before (Sokolovskiy et al., 2020), and typically it leads to a dipolar breaking for the upper layer vortex. Instead, we can see the formation of a tripole, also in the second layer. In the $3^{rd}$ and $4^{th}$ layers are formed small vortices that remain aligned with the initial vortex position. For longer times the vortexes continue to rotate until complete dissipation since there is no background





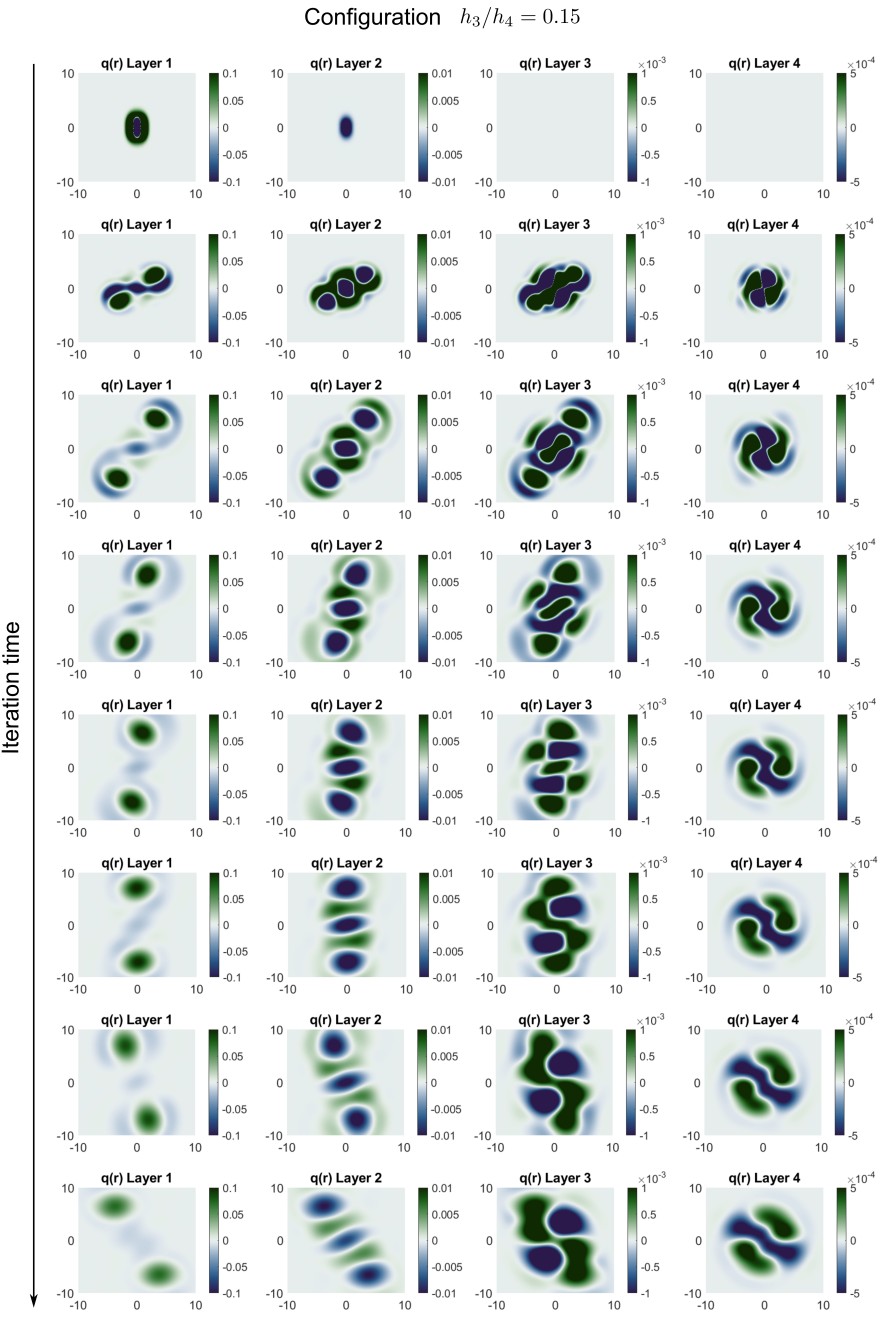

**Figure 12.** Time evolution of the spatial fields in the perturbed 4-layer case for a few iterations, in the case of a bottom layers thickness ratio of $h_3/h_4 = 0.15$. Columns are organized by layer, and rows by iteration time. The grid is zoomed to better capture the evolution shape. Also note that each layer colorbar has a consistent range in time but differs from layer to layer.







**Figure 13.** Time evolution of the spatial fields in the perturbed 4-layer case for a few iterations, in the case of a bottom layers thickness ratio of $h_3/h_4 = 0.77$. Columns are organized by layer, and rows by iteration time. The grid is zoomed to better capture the evolution shape. Also note that each layer colorbar has a consistent range in time but differs from layer to layer.





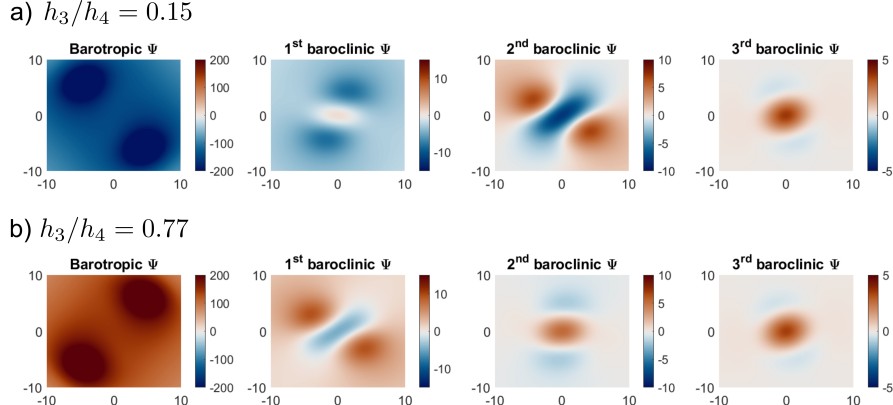

**Figure 14.** Streamfunction projection on the barotropic and baroclinic modes in the physical plane for (a) $h_3/h_4 = 0.15$ and (b) $h_3/h_4 = 0.77$ of the perturbed 4 layer case, for the last iteration showed in Figures 12 and 13. The colorbar ranges are different for different modes.

flow to interact with. The bottom layers concur to dissipation acting as sinks of potential vorticity, as can be seen clearly in
Figure 12 bottom panels intensification.

The most interesting outcome is the fact that the critical behavior observed for the stationary case is reproduced also in the first two layers when the perturbation is considered. In fact, for $h_3/h_4 = 0.77$, which was near the critical transition in the stationary case (Figure 10a), the rotation direction inverts also in the first layers (Figure 13). Moreover, the potential vorticity that reaches the bottom layers is way higher in this case than in the stationary case, thus comforting us on the fact that the
presence of a deep stratification can indeed have a non-negligible impact when considering more realistic evolution.

This is consistent with the barotropic modal structure of the stationary case (Figure 9), and can be observed in the correspondent modal projection for the perturbed case (Figure 14). The sign of the modes in the stationary and perturbed cases is the same, the perturbation changes however the modal structure, modulating the energy distribution. The presence of the perturbation highlights the strong connection of stratification configuration with the first two baroclinic modes (Figure 14)).
For the critical transition, not only the barotropic mode changes sign and orientation, but $1^{st}$ and $2^{nd}$ baroclinic modes behave very differently, and emerges a symmetry in the modal structure around the critical value. The $2^{nd}$ mode structure in Figure 14a is reproduced by the $1^{st}$ mode for the critical configuration (Figure 14b), while the $1^{st}$ mode in Figure 14a and the $2^{nd}$ in Figure 14b are completely different, in both shape and sign. The $3^{rd}$ mode appear unchanged, as in the stationary case.

Therefore, even when considering a regular perturbation, the mere presence of a dense deep layer, depending on its thickness
with respect to the upper one, can act as a stabilizer or as a disruptor on ocean vortices. This is also relevant in our case study area, where abyssal vortices exist and were observed by Rubino et al. (2012) in 2002-2003, when in situ data depict a relative stratification of $h_3/h_4 = 800/1500 = 0.53$. This value, as can be seen in Figure 5c, corresponds to a higher vorticity content in the $3^{rd}$ layer and a co-rotation of the $3^{rd}$ and $4^{th}$ layers, thus being one of the configurations ensuring the vortex stability condition discussed earlier (Benilov, 2003).





This contribution of deep stratification in deep sea variability we found, despite being small, can have important impacts when integrated over time in energy budgets. The deep sea varies with small scales compared to the rest of the ocean, but even small variation can have cascade impacts, as seen for example in the EMT, where a variation of $\sim 0.2°C$ in temperature and of $\sim 0.1\,PSU$ in salinity in the abyss (Artale et al., 2018) brought abrupt changes on the circulation patterns of the Mediterranean Sea.

## 5 Conclusions

We applied a quasi-geostrophic approach on a two-dimensional ocean, with four non-linearly coupled layers and observation-based parameters, to investigate the role that abyssal stratification can have on the propagation of potential vorticity. The propagation and stabilization of potential vorticity through the bottom depend on the relative thicknesses and stratification of the layers, which have proven to be critical factors in reshaping the water column circulation patterns. The dense and

stable deep layer observed in the Ionian Sea starting from the late 1990s gave an example of the baroclinic exchange from kinetic to potential energy along the vertical (Artale et al., 2018; Manca et al., 2006; Klein et al., 1999; Li and Tanhua, 2020; Giambenedetti et al., 2023). This is an example of a wider tendency of the global oceans, as recognized by the latest Intergovernmental Panel on Climate Change (IPCC) reports: ocean stratification has increased substantially over the past decades, even at great depths (Li et al., 2020).

The formulation of our model did not follow the usual layered models employed in studying vortex stability (Sokolovskiy, 1997; Carton et al., 2014; Katsman et al., 2003) or potential vorticity propagation from the surface (Smith and Vallis, 2001), because we were not focused on the evolution of a perturbed vortex on short time scales, nor in higher order processes like those involved in the upper layers. Instead, we wanted to investigate the problem with an upside-down point of view, trying to understand the impact on potential vorticity propagation of long time scale stratification changes in the abyss like those

observed in the available data. Hence, we used a time parametrization artifice analyzing a set of stationary states to study the non-linear system evolving over a decade.

The simplifications and assumptions made though proved to give a realistic, yet more qualitative than quantitative, representation when compared with different observations. In fact, the Rossby internal radii from our model are consistent with the Ionian Sea circulation trends, and the phase speed values estimated from available observations have the same order of

magnitude as that of our model (Figure 4). Moreover, the time parametrization of the stratification evolution was justified a posteriori by the finding of critical ratios of bottom layers thicknesses (Figure 5c).

We found that as the thickness of the bottom denser layer decreases, the potential vorticity signal initially tends to have the same rotation direction as the above layers, as long as it reaches a critical value of the ratio between the thickness of the last two layers below 1 where a sort of phase transition occurs. The potential vorticity signal thus increases in the bottom layer

and the rotation direction inverts. This critical behavior is not present when considering fewer layers (Figure 10), hinting at the fact that the usual picture of a two-layer Ionian found in literature may be is too simplified for a completed representation of all the processes of the ocean water column (Rubino et al., 2020; Gačić et al., 2021). Moreover, the impact of this critical



behavior extends to the surface layers, in fact the conversion between the first two baroclinic modes also alters the sign of the barotropic mode (Figure 9), and the effects of this instability on the upper layer is evident when perturbing the evolution of the vortex. In fact, we observe that near the critical thicknesses ratios values of the bottom layers found in the non-perturbed case, the rotation of the first layer inverts as well (Figures 12 and 13).

Therefore, regardless of stronger dissipation, forcing, background flow, and other instabilities one considers in a more accurate description of the flow dynamics, the net effect of the deep stratification variability is to reshape the structure of the mean circulation and mass distributions for the whole water column.

The stratification-induced inversion we found for some particular configurations of layers thickness is also consistent with local observations in the Ionian Sea, as can be seen for example in the hodographs in Figures 1a and b. The deep layers stratification in the Ionian Sea changed drastically for the EMT, during which the continuous entrainment of warmer and saltier Aegean waters created the preconditions for instabilities and heat transfer through diffusion and turbulent mixing. After a few decades, the mixing due to the concurrence of different properties (e.g. tides, IWs, topography/morphology, etc.) homogenized the bottom layer in the Ionian abyssal plain.

We demonstrated the crucial role of stratification in directing the rotation of potential vorticity generated at the interface between layers of different densities, and that the whole water column engages in the abyssal dynamics. This can have a role in stabilizing the propagation of Rossby waves, affecting in turn the sub-surface circulation. Further, it points out the impact of the bottom on the vertical propagation, which can have interesting implications on the redistribution of the energy stored by the deep sea, thus needing further investigation.

Our findings give a hint on how the deep variability is connected with the intermediate and surface dynamics through vertical stratification, and we suggest that this should be considered in more comprehensive models, particularly when considering topographic effects and their interaction with a more realistic stratification than the present work, as it can yield interesting outcomes.

*Code availability.* The code developed for this study is available at: https://github.com/bgiambe/QG4L.

*Data availability.* Original raw data of the CTD profiles were provided by Federico Falcini, Giorgio Riccobene, and Antonio Capone. The processed datasets used for this study is available at http://doi.org/10.5281/zenodo.7871735. NEMO-SN1 observatory data are available at http://moist.rm.ingv.it/.

*Author contributions.* BG, NLB, and VA designed the experiment and BG carried it out, developing the model code supervised by VA, and performing the simulations. BG prepared the manuscript with contributions from all co-authors.



*Competing interests.* The authors declare that they have no conflict of interest.

*Acknowledgements.* Thanks to Federico Falcini for providing the CTD data that help us in characterizing the water column and for the fruitful conversations. The NEMO-SN1 observatory data used are from the post-processed version carried on in the framework of the INGV (Istituto Nazionale di Geofisica e Vulcanologia) project MACMAP (A Multidisciplinary Analysis of Climate change indicators in the Mediterranean And Polar regions), that has in its objectives also the study of deep sea dynamics.

435





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
