# Peer review of "Semi-analytical approach to study the role of abyssal stratification in the propagation of potential vorticity in a four-layer ocean basin"

_EGUsphere, 2024_

## Referee Comment (RC1)

- The title is misleading: the approach is not semi-analytical. It is a numerical model. I am also uneasy with the notion of vertical propagation on PV when the latter is materially conserved in absence of diffusion and that there is no vertical advection in QG.

- I am uneasy with the justification of the equations using the depth $z$ instead of $\rho$, as the justification (based on crude finite difference) differs significantly from the rigorous derivation (based on vertical averaging, see e.g. [1]) of layered models. There must be other ways to justify the change (such as arguing that at the relevant order in $Ro$, isopycnals are flat - corrections only dynamically matter at higher order in $Ro$).

- line 145: the statement is technically erroneous. $\partial\psi/\partial z$ in equation (2) is a rescaled buoyancy anomaly. The vertical velocity is proportional to the material derivative of this buoyancy anomoly. So, although $\partial\psi/\partial z = 0$ implies no vertical velocity, the reciprocal is not true: no vertical velocity does not imply $\partial\psi/\partial z = 0$.

- Please clarify where equation (5) comes from.

- Equation (12) is unecessarily general since only $U_1 \neq 0$. It would be simpler to just define $\bar{\psi}_1$ in equation (12) and state in the text that $\bar{\psi}_j$, $j = 2, 3, 4$ is set to 0 at $t = 0$. There is no point in plotting curves $\bar{\psi}_j = 0$ in Fig. 3(a).

- The parameter $\beta$ appears in the equations but its value is not given (unless I am mistaken).

[1] V. Zeitlin, Geophysical Fluid Dynamics: understanding (almost) everything with rotating shallow water models, OUP, 2018

---

## Author Comment (AC1)

**RC1 Response**

Dear Referee #1,

Thank you for your thorough review of our manuscript. Your comments helped us understand a critical point in our manuscript that needs to be addressed to make our work better understood. We just realized that we had used ambiguous terminology that misled the interpretation of our intentions and methodology. We appreciate your feedback and have addressed your comment as follows:

- **Comment:** The title is misleading: the approach is not semi-analytical. It is a numerical model. I am also uneasy with the notion of vertical propagation on PV when the latter is materially conserved in absence of diffusion and that there is no vertical advection in QG.

  **Response:** We acknowledge the concern regarding the title and description of our approach. In the title and the text, we used the term propagation to give the idea of the transmission of PV signal throughout the water column, given the rearrangement due to the non-linear coupling. As you highlighted this term may result in a misunderstanding given the absence of vertical advection in the QG approximation, hence we clarified in the text. Since we do not use primitive equations to evolve and model the whole system and find a solution numerically for a limited case after deriving the governing equations for the system we think that "semi-analytical" is representative in our case. However, this may mislead in the title, given that we are presenting a toy model to investigate an observed process so we are changing the title to:

  "Study of the role of abyssal ocean stratification in the rearrangement of potential vorticity through the water column"

  and lines 5-7:

  "A quasi-geostrophic level model, with four coupled layers, a free surface, and a mathematical artifice for parametrizing decadal time evolution has been considered, proving that the relative thicknesses and the density difference among the layers are the two critical factors that determine the dynamical characteristics of this rearrangement."

- **Comment:** I am uneasy with the justification of the equations using the depth $z$ instead of $\rho$, as the justification (based on crude finite difference) differs significantly from the rigorous derivation (based on vertical averaging, see e.g. [1]) of layered models. There must be other ways to justify the change (such as arguing that at the relevant order in $Ro$, isopycnals are flat - corrections only dynamically matter at higher order in $Ro$).

  **Response:** This comment is pivotal for us to improve our discussion and better explain our motivations. As we tried to highlight in sections 1 and 2, our study was inspired by local observations. This led us naively to use interchangeably the terms "layer" (since

we are describing observed layers in the stratification of the Ionian Sea) and "level" (since our model is technically a level model and not a layered model) and we understand now that this confuses, instead of helping in the interpretation. In fact, using the depth z instead of ρ is a standard derivation for level models, which do not assume a linear relationship between the density and depth like in layer models which are designed to better follow isopycnals. As we explained in the text, given the abyssal density structure observed we cannot use a layer model approximation, and this is why we resolved to a level model. We revised our text using the correct terminology, avoiding this ambiguity that you helped us identify.

Relevant changes in the text:

Lines 51-56:

"Given the presence of the dense abyssal layer observed in the Ionian Sea, the best representation is depicted by a QG equation with four non-linearly coupled levels having parameters based on in-situ data. We used a level model instead of a layer model, meaning that the discretization uses a z vertical coordinate instead of a ρ coordinate to follow isopycnals. The two schemes are equivalent when ρ can be considered as a linear function of z, but this is not our case. In fact, the density structure observed in the Ionian Sea abyss is not linear in depth, hence we needed a coordinate that accounted for this variability through the entire water column. In numerical models of the ocean both approaches are combined, the vertical coordinate is typically used only in the surface layers, while the interior, which is considered more stable, is treated using isopycnals (Griffies et al., 2000)."

lines 128-136:

"Equation (2) contains derivatives in z, which must be discretized to conform with a 4−layer representation. We use a level model, i.e., the finite-difference discretization considers fixed vertical levels to define the layers instead of density. Since the discretization is performed on z directly, instead of taking ρ as the vertical variable, the formalization of the level model is slightly different than the more common layer models (Vallis, 2017; Cushman-Roisin and Beckers, 2011). However, the two approaches are equivalent when considering long-scale systems where density is a linear function of z, and are commonly combined in more comprehensive ocean models (Griffies et al., 2000). Using z as a vertical coordinate makes it possible to use the simplest discretization approach, accurately representing pressure gradients and equation of state for a Boussinesq fluid (Griffies et al., 2000)."

lines 376-379:

"We applied a quasi-geostrophic level model on a two-dimensional ocean, with four non-linearly coupled layers and parameters observation-based, to investigate the role that abyssal stratification can have on the rearrangement of potential vorticity through the water column. The transmission and stabilization of potential vorticity through the bottom depend on the relative thicknesses and stratification of the layers, which have proven to be critical factors in reshaping the water column circulation patterns."

lines 385-390:

"The formulation of our model did not follow the usual layer models employed in studying vortex stability (Sokolovskiy, 1997; Carton et al., 2014; Katsman et al., 2003) or potential vorticity propagation from the surface (Smith and Vallis, 2001; Zeitlin, 2018), because we were not focused on the evolution of a perturbed vortex on short time scales, nor in higher order processes like those involved in the upper layers. Instead, we wanted to investigate the problem with an upside-down point of view, trying to understand the impact on the potential vorticity structure in the water column due to long time scale stratification changes in the abyss, like those observed in the available data."

- **Comment:** line 145: the statement is technically erroneous. $\partial\psi/\partial z$ in equation (2) is a rescaled buoyancy anomaly. The vertical velocity is proportional to the material derivative of this buoyancy anomaly. So, although $\partial\psi/\partial z = 0$ implies no vertical velocity, the reciprocal is not true: no vertical velocity does not imply $\partial\psi/\partial z = 0$.
  **Response:** We apologize for the oversight. Our intention was to keep the mathematical notation as straightforward as possible to enhance readability and accessibility, but unfortunately, we oversimplified it. We did not mention that we applied the method of separation of variables to the streamfunction to apply the boundary conditions, which is a commonly used but important assumption to follow the mathematical passages. To clarify this, we added a more detailed derivation and explanation.

  Relevant changes in the text:

  lines 142-165:

  "To apply the boundary conditions in the vertical we employed the variable separation method for the streamfunction, i.e., we assume that $\psi(x,y,z, t) = \tilde{\psi}(x,y, t)\psi'(z)$. " and we changed eq. (4), (5), (6), and (7) accordingly.

- **Comment:** Please clarify where equation (5) comes from.

  **Response:** As for the previous comment, we added a more detailed derivation and explanation of the boundary conditions application.

- **Comment:** Equation (12) is unnecessarily general since only $U_1 \neq 0$. It would be simpler to just define $\bar{\psi}_1$ in equation (12) and state in the text that $\bar{\psi}_j$, $j = 2,3,4$ is set to 0 at $t = 0$. There is no point in plotting curves $\bar{\psi}_j = 0$ in Fig. 3(a).
  **Response:** We agree with the suggestion on equation (12), hence we modified it accordingly. For what concerns the figure, we found it helpful to have a visualization of the activation of the bottom levels in the final state starting from zero, so if you agree we would like to keep Fig. 3(a) as it is for completeness.

- **Comment:** The parameter $\beta$ appears in the equations but its value is not given (unless I am mistaken).

**Response:** The $\beta$ was kept in the derivation of the scheme but in this work is not used, since in our case the $\beta$-effect is negligible, we clarified this in the manuscript and dropped the term in eq. (8).

We hope that these revisions address your concerns satisfactorily. Thank you again for your insightful comments, we think that it helped us improve the manuscript significantly. We uploaded the revised manuscript with highlighted corrections, and remain at your disposal for any further suggestions and discussion.

Best regards,

Beatrice Giambenedetti, Nadia Lo Bue, and Vincenzo Artale

---

## Author Comment (AC3)

**RC2 Response**

Dear Referee #2,

We greatly appreciate the thorough review and constructive feedback on our manuscript. We have carefully considered your comments and made several revisions to enhance the clarity and depth of our study. Below is a detailed response to each of your points:

**\*General comments:**

This study examines the abyssal circulation of the Ionian Sea, specifically how vertical coupling between surface, intermediate, and abyssal layers might produce some observed decadal-scale changes in the circulation. The approach is based on sets of four-layer quasigeostrophic models with different stratifications (layer thicknesses) initialized with a Gaussian vortex. Some of the model results are qualitatively consistent with features found in current meter data. The results are very sensitive to the prescribed stratification, and this sensitivity is examined.

The text reads generally well, and most figures are clear. I think there are two major points (detailed below) and several other specific ones that need to be addressed.

I think a central issue in this work is that there is important sensitivity to the choices of layer thicknesses (and densities), as thoroughly examined in the manuscript. This is expected in QG systems with a few layers. Because the primary motivation of the manuscript is to understand decadal changes in the abyssal circulation with the simplest possible model, it would make sense to have experiments with more layers representing the observed density profile (Figure 2) more accurately. This would better approximate the continuously-stratified limit (see, e.g., Gulliver & Radko's 2022 Figure 4 with a ten-layer model based on an observed profile), and help determine at which point the discrete representation of the stratification is sufficiently realistic to reproduce the observed processes.

**Response:** We acknowledge the importance of sensitivity to layer thicknesses and densities in QG systems, this was one of the motivations that led us to use this approach. We think that we may have used confusing terminology, leading to a misunderstanding of the method we employed, which was a level model in our case instead of a layer model (as the one employed in Gulliver & Radko's 2022 for example). We retained the term "layers" thinking naively that it would help with the interpretation since we are describing observed layers in the stratification of the Ionian Sea, but we revised the manuscript, clarifying when necessary that we are using "levels". The choice to use a level model, i.e. using the depth z instead of ρ without having to assume a linear relationship between the density and depth, and the number of levels, as explained in the text, was based on the identification of the water masses in the Ionian Sea, so as to reproduce the stratification observed with a sufficient number of levels to realistically represent the stratification while containing the computational cost. Though our study was motivated by observations in the Ionian Sea, this approach has a more general applicability being representative of ocean basins with intermediate and deep water inputs (following Vallis, 2017). The reference mentioned, which is really interesting and we think adds an important element to our brief discussion on the stability and which we have included now in the revised manuscript, uses a quite different approach, assuming constant density differences among interfaces, which is not realistic to our observations. Also, it solves in Fourier space as another cited work (e.g., Yassin, and Griffies, 2022), which is lighter in terms of computational cost with respect to our that solves directly in time, however, it would filter out some signals that we would like to include and changes the resolution of the final results, in particular when approaching the intermediate/deep water interaction and hydrological differences. However, we believe this approach is useful in other contexts as well and plan to employ it in the future to better understand our data, for example for a continuous stratification, incorporating more realistic forcings and boundary conditions.

Relevant                          changes                         in                           the                            text:
lines                                                         55-                                                             61:
"Given the presence of the dense abyssal layer observed in the Ionian Sea, the best representation is depicted by a QG equation with four non-linearly coupled levels having parameters based on in-situ data. We used a level model instead of a layer model, meaning that the discretization uses a z vertical coordinate instead of a ρ coordinate to follow isopycnals. The two schemes are equivalent when ρ can be considered as a linear function of z, but this is not our case. In fact, the density structure observed in the Ionian Sea abyss is not

linear in depth, hence we needed a coordinate that accounted for this variability through the entire water column. In numerical models of the ocean both approaches are combined, the vertical coordinate is typically used only in the surface layers, while the interior, which is considered more stable, is treated using isopycnals (Griffies et al., 2000). "

lines 387-390:

"We applied a quasi-geostrophic level model on a two-dimensional ocean, with four non-linearly coupled layers and parameters observation-based, to investigate the role that abyssal stratification can have on the rearrangement of potential vorticity through the water column. The transmission and stabilization of potential vorticity through the bottom depend on the relative thicknesses and stratification of the layers, which have proven to be critical factors in reshaping the water column circulation patterns."

lines 396-401:

"The formulation of our model did not follow the usual layer models employed in studying vortex stability (Sokolovskiy, 1997; Carton et al., 2014; Katsman et al., 2003) or potential vorticity propagation from the surface (Smith and Vallis, 2001; Zeitlin, 2018), because we were not focused on the evolution of a perturbed vortex on short time scales, nor in higher order processes like those involved in the upper layers. Instead, we wanted to investigate the problem with an upside-down point of view, trying to understand the impact on the potential vorticity structure in the water column due to long time scale stratification changes in the abyss, like those observed in the available data."

Another major point is the choice of a single vortex as initial condition. If the motivation is to explain some of the changes in the observed abyssal currents (Figure 1), decaying turbulence experiments with random broadband initial conditions would be more relevant. Even considering that the Intermediate Water eddies found in the Mediterranean have a Gaussian velocity profile, It seems unlikely that changes in individual eddy structure and propagation could be responsible for the observed changes in a real, broad-banded flow with a developed turbulent cascade. Diagnosing the changes in the baroclinic/barotropic energy fluxes in a turbulent four-layer system (with different stratifications like the authors do) could therefore be more helpful to find quantitative links with the observations.

**Response:** We agree that decaying turbulence experiments with random broadband initial conditions could offer more relevant insights into the local variability of the abyssal currents, in fact, we addressed some of this aspect in our precedent work (Giambenedetti et al., 2023) where we also recognized the impact of the area's topography and which we refer to in the paper. However, this study's primary focus is on the impact of individual eddy structures on decadal changes in abyssal currents, so the temporal scales involved are quite different. Future work will aim to explicitly address these dynamics through more complex simulations, so thank you for your suggestions which we are incorporating in a new manuscript.

**\*Specific comments:**

Figure 1c,d and paragraph starting at line 84: As seen in the hodographs and noted in the text, the amplitudes of the mean flow are similar, but there seems to be much less energy in the subinertial band in the SMO-1 rotary spectra than in GNDT-1. Is this low frequency/mesoscale kinetic energy drop reported in other observations in the literature, and if so, is the reason understood?

**Response**: Thank you for this comment, this opens a wider discussion. We think that the signal observed in the Near Inertial band by the seafloor observatory deserves a more thorough study, which we are currently working on. Our idea is that the difference in stratification between the two periods played a key role in allowing wave-wave interactions (there is a small but notable peak in correspondence of the frequency f-M2 only in SMO-1 rotary spectra), but it is still a work in progress, that however are revealing a presence of vorticity anomaly at the deep depth. In that area, there is also a strong influence of the Sicily escarpment, with its steep slope being heavily affected by active canyons, as well as the influence of the deep water that originated from the Adriatic basin. So, it is a complex topic that needs dedicated study.

Line 47: I think it is important to note here that adding bottom topography with realistic roughness does produce more stable vortices with much longer lifetimes (across different density stratifications), as often observed in real vortices (Gulliver & Radko, 2022).

**Response**: Thanks for this suggestion. We have revised the manuscript to highlight the role of bottom topography in stabilizing vortices, following Gulliver & Radko (2022). This aligns with our ideas on the factors influencing abyssal dynamics and energy redistribution and adds another piece to the puzzle of the energy budgets in the ocean.

Relevant changes:

lines 47-51:

"One proposed explanation for this longevity is the stabilizing effect of bottom topography by Gulliver and Radko (2022), which is an important effect often neglected. As suggested in Giambenedetti et al. (2023), the presence of a stable deep stratification can enhance topographic effects. Particularly when considering abyssal dynamics, considering topography alone leaves open questions about the connection between different layers along the water column."

Lines 113-114: It is said here that compressibility effects were corrected for in the stratification frequency, so I think it would make more sense to have potential density in Figure 2b and in the layer density values reported. Using in situ density rather than potential density is very unusual, and I do not see a case for this choice here. Because only lateral density gradients matter in the QGPV equations, the extra density term due to compressibility effects should make no difference dynamically (apart from any potential numerical error propagation). But I still see no reason to use in situ density instead, especially since it obscures vertical structure in the observed profile (Figure 2b).

**Response**: Potential density is referenced to a specific depth. Since we are considering the whole water column at different levels, it seemed more appropriate for our case to use in-situ density instead of having differently calculated densities for each level of the model.

Lines 226-234: The relationship between the interfacial deformation radii in a layered QG model (as the one in this work and in Carton et al., 2014) and the modal deformation radii in a continuously stratified system (as in, e.g., Nittis et al. 1993) needs to be clarified here. This is important because the layer-wise deformation radii increase towards the bottom, and that seems to agree with the scale of the circulation features. But the modal deformation radii decrease with increasing mode number, and are associated with vertical modes with more zero crossings (i.e., shorter vertical wavelength). The modal analogues of the layered deformation radii involve combinations of the layer thicknesses (e.g., sqrt(g' * H1*H2/(H1 + H2))/f for the baroclinic mode in a two-layer model). It seems that the abyssal gyres in question are better described in the layered sense as locally equivalent-barotropic features, so I suggest trying to clarify that point.

**Response**: Thank you for this insightful comment. To clarify this point, we have revised the relevant section as follows (lines 230-238):

"In layered QG models, the interfacial deformation radii are determined by layer-wise stratification parameters. Thus, using level thicknesses, Coriolis parameter, and densities is possible to evaluate the interfacial internal deformation radius $L\_j\_d =sqrt(g'\_j h\_j)/f$ for the j −th level, which is a baroclinic Rossby radius of deformation, taking into account the reduced gravity and relative thickness of the levels (LeBlond and Mysak, 1978). In continuously stratified systems, the modal deformation radii decrease with increasing mode number (Nittis et al., 1993). Higher modes are associated with shorter vertical wavelengths and more zero crossings in their vertical structure functions. On the other hand, in a layered model, the radii tend to increase towards the bottom of the water column because of the reduced gravity, and the thickness of the layers changes with depth. This is easily explained in a two-layer model (LeBlond and Mysak, 1978; Carton et al., 2014), but in our case, it depends on the non-linear coupling among the different levels, resulting in a locally equivalent-barotropic dynamic."

Figure 5: The axis labels are very difficult to read without zooming in, the font sizes need to be increased.

**Response**: The font sizes of the axis labels in Figure 5 have been increased to enhance readability.

**\*Minor corrections and editorial suggestions**

**Response**: We revised our manuscript following your suggestions and rewrote the confusing parts to clarify the text:

lines 353-355:

"The bottom levels contribute to dissipation by acting as sinks of potential vorticity, as clearly shown by the intensification in the bottom panels of Figure 12."

Thank you for your thorough review and your valuable input. Your suggestions have not only enhanced the clarity and depth of our current work but have also highlighted important areas for future research that we are eager to explore. We remain at your disposal for any further suggestions and discussion.

Sincerely,

Beatrice Giambenedetti, Nadia Lo Bue, and Vincenzo Artale

---

## Author Response (AR2)

Dear Anne Marie Treguier,

Thank you for your kind words and the update regarding the manuscript. We changed the axis labels in Figure 5 as suggested.
We appreciate the support of Ocean Science in this process and look forward to publishing our work.

Best regards,

Beatrice Giambenedetti, Nadia Lo Bue, Vincenzo Artale